# Reducing Weaning Stress in Piglets by Pre-Weaning Socialization and Gradual Separation from the Sow: A Review

**DOI:** 10.3390/ani13101644

**Published:** 2023-05-15

**Authors:** Céline Van Kerschaver, Diana Turpin, Joris Michiels, John Pluske

**Affiliations:** 1Laboratory for Animal Nutrition and Animal Product Quality, Department of Animal Sciences and Aquatic Ecology, Ghent University, Coupure Links 653, 9000 Ghent, Belgium; 2Department of Primary Industries and Regional Development, 3 Baron-Hay Court, South Perth 6151, Australia; 3Australasian Pork Research Institute Limited, Willaston 5118, Australia; 4Faculty of Science, The University of Melbourne, Parkville 3010, Australia

**Keywords:** piglet, weaning, stress, socialization, co-mingling, maternal separation, intermittent suckling

## Abstract

**Simple Summary:**

In commercial pork production, piglets are weaned at a fairly young age. Piglets are removed from the sow and are subject to sudden changes in their diet, environmental conditions and social grouping. For the piglet, this causes major upheaval and disruption to eating and behavioral patterns, leading to distress, gastrointestinal tract dysfunction and behavioral disorders after weaning. This contrasts with natural conditions where nest-leaving of the sow and her offspring and joining the herd with increasing interactions with non-littermate piglets occurs between approximately 6.5 and 15 days after birth, with weaning typically completed between 14 and 18 weeks of age. Strategies that could be adapted to a commercial setting and that allow socialization of non-littermate piglets before weaning and gradual separation of the litter from the sow in the period leading up to weaning may reduce the stress occurring after weaning and improve performance, welfare and gastrointestinal function. This review summarizes current knowledge concerning such strategies.

**Abstract:**

The weaning of pigs in most commercial pork production systems is an abrupt event performed at a fairly young age, i.e., mostly between 2.5 and 5 weeks of age. This practice induces a stress response, and its impact on behavior, performance and the gastrointestinal tract has been well described. Historically, there has been a focus on pre- and post-weaning nutritional strategies and post-weaning housing conditions and medication to improve production and reduce mortality after weaning. However, alternative pre-weaning housing and management systems that promote the development of natural social behaviors of piglets before weaning have recently received more attention. Co-mingling of non-littermates before weaning is a strategy that aims to initiate social interactions prior to weaning. The separation of the litter from the sow in the period leading up to weaning, termed intermittent suckling, aims to enhance the gradual separation from the sow. In addition, these practices encourage the young pig to learn explorative nutrient sourcing. Altogether, they may reduce weaning-associated stress. In this review, these strategies are defined, and their effects on behavior, performance, mortality, gastrointestinal function and immunocompetence are described. Though these strategies may be adapted to a commercial setting, it also becomes clear that many factors can contribute to the success of these strategies.

## 1. Introduction

The weaning of piglets in most commercial pork production systems worldwide is an abrupt event and is performed at a fairly young age. Piglets are simultaneously removed from maternal care, mixed and moved into a new environment, and offered solid feed devoid of the antibodies and other gut protective factors and stimulants found in sow’s milk. This series of events causes an upregulation of the hypothalamic-pituitary-adrenal (HPA) axis and, accordingly, an acute increase in cortisol levels, irrespective of weaning age, indicative of physiological and psychological stress [1]. The weaning-associated stress response and its impact on performance and the gastrointestinal tract (GIT) have been well described [2,3,4,5,6]. Historically, there has been a focus largely on pre- and post-weaning nutritional strategies and post-weaning housing conditions and medication to improve production and reduce morbidity and mortality after weaning [7,8,9]. However, alternative pre-weaning housing and management systems that promote the development of natural social behaviors in piglets before weaning have recently received more attention for their potential to ease the weaning transition and improve performance and health.

With this in mind, the general aim of this review was to summarize the literature related to two techniques, namely co-mingling of non-littermates before weaning and intermittent suckling (IS), on the effects of weaning-associated stress and post-weaning performance. Co-mingling of non-littermates before weaning is a strategy that aims to initiate social interactions prior to weaning. The separation of the litter from the sow in the period leading up to weaning, termed IS, aims to enhance the gradual separation from the sow. In addition, these practices encourage the young pig to learn explorative nutrient sourcing, which may better prepare it for weaning. Together, they may reduce weaning-associated stress. The factors that contribute to the success of these strategies will be discussed in detail. In order to fully understand how these two pre-weaning techniques may affect the pig and its response upon weaning, an in-depth understanding of the development of social and eating behaviors of pigs in (semi-)natural conditions and of the effects of commercial weaning practices on piglets’ behavior and stress response is warranted. Therefore, first, the related literature is summarized in the two following sections.

## 2. Development of Social and Eating Behaviors of Pigs in (Semi-)Natural Conditions

In the late 20th century, an elegant series of studies was conducted to provide insights into the social and eating behaviors of domestic pigs when kept in (semi-)natural conditions (Table 1). The weaning-associated behaviors described in these studies are believed to be very similar to those of wild pigs. The following section describes these studies’ major findings and possible implications for piglets raised commercially.

### 2.1. Nest Occupation and Maternal Care

Sows isolate themselves from the herd to build a nest, to farrow and to care for their young [10]. After birth, the first interaction between the sow and her offspring, i.e., nose-to-nose contact and sniffing, occurs before piglets suck at the teat for the first time and serves to quickly establish the maternal bond. This first contact takes place earlier when it is initiated by the sow than when initiated by the piglet (20.2 vs. 35.7 min post-partum) [11]. This selective bonding allows the mother and her young to quickly identify each other and is crucial to ensuring maternal care and survival [12]. Newborn piglets have functional sensorial systems (e.g., auditory and visual) for early learning, and, through neurobiological mechanisms and neurotransmitters, behavioral preference for filial individuals is established. Although newborn piglets have a high degree of locomotory development, they will remain in the nest for the days to come [13]. During the first day after farrowing, more than 85% of all sucklings are initiated [14,15] and less than 5% of all sucklings are terminated [15] by the sow. The latter occurs to prevent fights at the udder due to competition for teats between littermates [16]. The synchronization of suckling bouts is already developed during colostrum provision, and complete suckling synchronization is achieved by day 3 [17]. The sow initiates suckling by grunting, approaching and nosing her piglets, after which the piglets awaken, approach the udder and start sucking [16]. Within 3 days of birth, piglets will develop a teat preference [11]. Piglets sniff the udder and sample different teats, and fighting between littermates is often involved [16]. Piglets stay in the nest during the first three days after birth, only leaving the nest to urinate and defecate [18]. From day 4 post-partum, piglets follow the dam outside the nest, but they still remain close to her. As piglets age, their activity increases and they follow their mother over larger distances [18,19]. Hence, the first encounters with other members of the herd take place. These approaches are mainly nose-to-nose contact and occur peacefully [20].

### 2.2. Social Integration within the Herd

Between approximately 6.5 and 15 days after birth, the sow leaves the nest together with her offspring [10] and occupies a new sleeping ground every night in proximity to the herd [21], finally joining the herd at about 9 days post-partum [15,18]. Approximately 10 days after nest leaving (up to week 8), piglets spend less time at the udder (on average 17% of the observation time) and undertake other activities more, such as walking and standing (on average 21% of the observation time) [22]. When pigs become older, they need more food than the sow can provide or is willing to supply. Therefore, piglets try to initiate suckling by approaching the udder, whining, sucking and massaging the teats. However, the mother still determines the precise timing of suckling [16], and sucklings are still ended by the sow. Jensen (1988) [23] concluded that the percentage of sucklings terminated by the sow significantly increased between 1 and 4 weeks post-partum. For example, Jensen et al. (1991) [15] showed that, on day 1 post-partum, less than 5% of the sucklings were terminated by the sow, whereas this was about 60% on day 10 post-partum. The ratio of time spent at the udder versus other activities starts to change. Piglets are more confident in exploring greater distances from the sow, which presents more opportunities for social contact with non-littermates. This is especially evident during sow feeding events. Between days 7 and 18 after farrowing, the piglets attend, with their mother, for the first time, the feeding sites [10,20] and then continue to do this every morning. When the sow is eating with other group members, piglets are resting at a distance of 10–20 m from the feeding site, where the occasion for social interaction with other group members again increases [20]. Since this feeding takes place in a small area, the likelihood of social interaction with non-littermates increases accordingly. Hence, between weeks 2 and 7 post-partum, social interactions between piglets and other members of the group are numerous and frequent. It seems that, after week 7, the process of social integration, which can be defined as the process whereby pigs become accepted and form stable social bonds within the herd, is completed. According to Petersen et al. (1989) [20], social interactions between weeks 2 and 5 mainly occur amongst littermates, whereas between 8 and 12 weeks, interactions with non-littermates and other members of the group become as important. In addition, this study demonstrated that 60.9% of all observed social interactions amongst piglets and other animals between weeks 2 and 17 post-partum consisted of nose-to-nose contact. Only 17.3% of all interactions were considered aggressive, i.e., head knocking, shoulder-to-shoulder movements and neck biting. As piglets mature, their activity increases, but the frequency of social interactions decreases. Petersen et al. (1989) [20] showed that, in week 2 after farrowing, there were 25.7 interactions per hour, while in weeks 7, 8–12 and 15–17, it declined to 5.3, 5.6 and 3.9 interactions per hour, respectively. During this time, piglets also start to explore foods other than milk. For example, between days 24 and 36 after farrowing, piglets start grazing and eating at the feeding place [20].

### 2.3. Slow and Gradual Weaning

Piglets are weaned naturally between 14 and 18 weeks of age, according to observations by Jensen (1986) [10], Jensen and Recén (1989) [24], Jensen and Stangel (1992) [25], Newberry and Wood-Gush (1985) [16] and Petersen et al. (1989) [20]. The weaning process of piglets in (semi-)natural conditions is slow and gradual. Suckling-related activities slowly decline and consequently disappear when the weaning process is completed [23,24], in part caused by changes in dam behavior. Jensen (1988) [23] observed that the proportion of sucklings initiated by the sow decreased significantly between weeks 4 and 10 post-partum. From week 10 post-partum, the decrease in the number of sucklings is even more pronounced, further stimulating the piglets to feed themselves with sources other than milk. Jensen (1986) [10] and Jensen and Recén (1989) [24] indicated that weaning involves a rapid decline or atrophy of the udder of the sow. When there was no suckling for two consecutive days, they observed a clear degeneration of the udder 2–3 days afterwards [24]. The weakening of the family bond between the sow and her litter is also an indication of the weaning process [25]. To our knowledge, no study has looked at the stress response under (semi-)natural conditions of weaning, but it is probable that pigs may not experience a growth check during this period [26], as observed following abrupt weaning as practiced commercially.

**Table 1 animals-13-01644-t001:** Literature overview of characteristics of pigs in (semi-)natural conditions.

Reference	Breed	Group Size and Composition	Environment	Distance from Farrowing Nest to the Group (m)	Litter Size (Mean and [Range])	Weaning Age (Mean and [Range]) (Weeks Post-Partum)
Castrén et al., 1989 [14]	Swedish Landrace	- Group 1: 4 gilts - Groups 2 and 3: 1 old boar, 4 adult sows and offspring, 4 gilts	Tovetorp Research Station, Stockholm, Sweden 3 adjacent enclosures Enclosure 1: group 1, primary forest, 1 ha Enclosures 2 and 3: groups 2 and 3, several biotopes: open field, swamps, mossy areas, primary forest, large fir plantations, 7 and 13 ha		[5–13]	
Jensen, 1986 [10]	Swedish Landrace	- Group 1: 1 boar, 2 gilts - Group 2: 1 boar, 1 sow and offspring, 2 gilts	Tovetorp Research Station, Stockholm, Sweden Hilly area with few steep slopes 2 adjacent enclosures: several different biotopes, comprising open fields, swamps, mossy areas, primary forest, large fir plantations, 7 and 13 ha	225–550		[14–17]
Jensen, 1988 [23]	Swedish Landrace	2 groups each consisting of 1 older and 1 younger boar, 3–4 adult sows and offspring, 4 gilts	The same as in the study of Jensen (1986) [10]		7.5	
Jensen et al., 1987 [27]	Swedish Landrace	2 groups each consisted of 1 older and 1 younger boar, 3–4 adult sows and offspring, 4 gilts	The same as in the study of Jensen (1986) [10]			
Jensen et al., 1991 [15]	Swedish Landrace	- Group 1: 1 boar, 4 sows and offspring, 4 gilts - Group 2: 1 older and 1 younger boar, 4 adult sows and offspring, 4 gilts	The same as in the study of Jensen (1986) [10]			
Jensen and Recén, 1989 [24]	Swedish Landrace	2 groups each consisting of 5–10 sows and offspring, 1–2 adult boars	The same as in the study of Jensen (1986) [10]		[7–10]	17.2 [15.6–19.5]
Jensen and Redbo, 1987 [21]	Swedish Landrace	2 groups each consisting of 1 old and 1 young boar, 2–3 adult sows and offspring, 4 gilts	The same as in the study of Jensen (1986) [10]	150–250		
Jensen and Stangel, 1992 [25]	Swedish Landrace	1 boar, 5–8 sows and offspring, 0–4 gilts	Hilly area Enclosure contains several biotopes: pastures, swamps, forests, large fir plantations, small streams and other sources of natural water Different shelters			18.9 [15–22]
Newberry and Wood-Gush, 1985 [16]	Large White	1 adult boar, 5 adult sows and offspring, 10 juveniles from previous litters	Pig Park, Midlothian, Scotland Hilly area Enclosure contains running streams, open grassy areas, gorse-covered areas and pine trees			[8.6–14.3]
Newberry and Wood-Gush, 1986 [19]	Large White	1 boar, 5 sows and offspring	The same as in the study of Newberry and Wood-Gush (1985) [16]			
Petersen, 1994 [22]	Swedish Landrace	Year 1 - Group 1: 1 boar, 5 sows, 1 gilt - Group 2: 2 boars, 4 adult sows, 4 gilts Year 2 2 groups each consisting of 1 boar, 4 sows, 4 gilts	The same as in the study of Jensen (1986) [10]			
Petersen et al., 1989 [20]	Swedish Landrace	- Group 1 2 sows, 1 older and 1 younger boar, 4 gilts - Group 2 3 sows and offspring, 1 older and 1 younger boar, 4 gilts	The same as in the study of Jensen (1986) [10]			[15–17]
Petersen et al., 1990 [11]	Swedish Landrace	- Group 1: 1 boar, 5 sows, 1 gilt - Group 2: 2 boars, 4 sows, 4 gilts	The same as in the study of Jensen (1986) [10]		10.2 [6–17]	
Stangel and Jensen, 1991 [18]	Swedish Landrace	- Group 1 1 boar, 4 sows and offspring, 4 gilts - Group 2 1 older and 1 younger boar, 4 sows, 4 gilts	The same as in the study of Jensen (1986) [10]	50–100		

## 3. Effects of Commercial Weaning Practices on Piglet Behavior and Stress Responses

### 3.1. Factors Leading to Weaning Distress and Post-Weaning Piglet Behavior

In contrast to the slow and gradual nature of the weaning process in (semi-)natural environments, under most commercial conditions worldwide, piglets are typically weaned abruptly between 2.5 and 5 weeks of age. This weaning practice is, therefore, undoubtedly a critical phase in the pigs’ lives that causes distress and behavioral disorders [28]. It is well accepted that the sudden change in nutrition and the pig’s social and physical environment is associated with physiological and psychological stress. It is clear that different stressors culminate upon weaning (reviewed by [29]), but to what extent single factors and their interactions contribute to the upregulation of the stress response and subsequent productivity is far less clear. Funderburke and Seerley (1990) [30] reported that plasma cortisol levels were higher in pigs that faced nutritional changes (from milk to dry feed) as compared to psychological (removal of sow) or cold stress (from 13 °C and below), which may suggest that the dietary switch had the greatest influence. In this study, though, piglets remained in the farrowing pen in all treatments, with only the sow being moved. When piglets are mixed and moved to the nursery, as occurs mostly commercially, the relative contributions of nutritional, psychological and environmental stressors to the post-weaning malaise are more difficult to establish. As has been described many times, a major consequence of weaning is a reduction in voluntary feed intake [5,31]. Common strategies to ease the dietary switch, therefore, are to provide solid creep feed or milk liquid feed during the suckling period and to offer the same feed(s) in the days following weaning. Creep feeding can be an effective strategy to familiarize piglets with solid food [7,32]. However, the intake of solid feed during lactation is generally small and highly variable within and between litters and may only be significant if weaned later than 3 weeks of age [33,34,35].

More relevant to this review is the effect of maternal separation. Its contribution to post-weaning stress in piglets cannot be ignored. Maternal separation is a well-characterized model of early-life stress in rodents. For example, Gareau et al. (2006) [36] demonstrated immediate transient effects on colonic mucosal barrier function and long-lasting effects on bacterial–host interactions in separated neonatal rat pups. Repeated 2 h daily isolation from days 3 to 11 of age in piglets caused changes in behavioral, neuroendocrine and immune regulation and produced long-term effects, not only on the activity of the HPA axis but also on the immune–brain circuitry [37]. Early separation from the mother is often manifested by prolonged vocalizations. Colson et al. (2006) [38] showed an increase in vocalizations and lying in litter cohesion, which was later followed by increased aggressive and (belly) nosing behavior in piglets weaned either at 21 or 28 days of age, and yet more intense in the first group. These behaviors could also be related to anorexia following weaning. However, low grunts and calls during several days after the weaning event could be attributed to pigs searching for their mother.

At weaning, piglets from different litters are usually brought together in larger groups, after which they are redistributed across different pens, often based on body weight and sex. Stress caused by grouping piglets has short-term and long-term consequences [39]. In the short term, the mixing of non-littermates induces an increase in body temperature and fighting behavior [39]. Piglets fight in order to impose dominance [40]. Ekkel et al. (1995) [41] and Pluske and Williams (1996) [42] demonstrated that piglets that are mixed at weaning show significantly more agnostic behaviors during the first hour after mixing than those that are not mixed at weaning. Moreover, it has been demonstrated that the mixing of litters at weaning not only intensifies interactions between unfamiliar piglets but also the recurrence of agnostic interactions between littermates [43]. Therefore, the sudden mixing of unfamiliar piglets results in high numbers of skin lesions and scratches. Bohnenkamp et al. (2013) [44] found that grouping and redistribution of non-littermates at weaning based on weaning weight may influence the agonistic behavior of piglets after weaning. They indicated that piglets with a weaning weight of 8 kg showed more agonistic behavior in the 40 h after weaning than piglets with a weaning weight of 6 or 10 kg. Mei et al. (2016) [45] found that, in a group where three pigs from one litter were mixed with three pigs from another litter, more fights occurred during the 5 h after mixing compared with the group where four pigs from one litter were mixed with two pigs from another litter, or where no piglets were mixed (respectively, 25.3 ± 4.8; 15.3 ± 2.2; 2.0 ± 0.5 fights). In addition, the time spent fighting was higher in the first group. Similarly, the number of aggressive interactions was higher in groups with four piglets from three different litters compared to groups with six piglets from two litters [46]. Blackshaw et al. (1987) [47] reported that mixing piglets from multiple litters caused severe agonistic behaviors during the 1.5 h after mixing. Moreover, they observed that mixing piglets from three litters at weaning caused less agonistic behavior during the 1.5 h after mixing, which is an important period for the establishment of a hierarchy, than mixing piglets from two or four litters. Interestingly, a recent study suggests that the frequency of agnostic interactions showed high fluctuation across groups of mixed weaners, yet for most pens, stable social networks were established by the third day after weaning [48]. Aggressive behavior caused by grouping unacquainted litters is associated with activation of the HPA axis and an increase in plasma cortisol concentrations [49,50,51]. The studies by Colson et al. (2012) [49] and Merlot et al. (2004) [51] demonstrated that social stress at weaning also causes a higher occurrence of lying or resting behaviors. Colson et al. (2012) [49] found that combining social and housing changes at weaning resulted in an increased behavior of the pigs lying inactive in an awake state during the 7 h after weaning. Moreover, the study by Merlot et al. (2004) [51] reported that the effect on cortisol and behavior is influenced by the social rank of the pig within the group. In the study, a dominance index was assigned to each piglet in the pen by observing the agonistic behavior of the piglets after grouping. The behavioral responses of dominant piglets differed from the behavioral responses of subordinate piglets. The researchers suggested that mixing piglets causes stress but that piglets can adjust to stressful situations by developing behavioral strategies. In the long term, the social stress caused by mixing unfamiliar piglets at weaning causes immunological and physiological disorders [52] and reduced performance of the animal [41]. For example, in the study by Ekkel et al. (1995) [41], the average daily gain of piglets that were mixed at weaning was lower compared to piglets that were not mixed or were moved to another pen at weaning (479 ± 12 vs. 528 ± 14 g/d). The effect of weaning age on post-weaning behavior has been well acknowledged [53]. The detrimental effects are more substantial when weaned earlier [53,54]. For example, a higher level of aberrant behaviors, such as belly nosing, vocalizations and escape attempts, was noticed in 7-day-old weaned pigs as compared to weaning at 14 or 28 days of age [53].

### 3.2. Weaning-Induced Stress Responses

Several neuroendocrine responses occur during this weaning-induced stress response via the sympathetic nervous system activation of the adrenal medullae and the HPA axis. The locus coeruleus/norepinephrine system in the brain releases the neurotransmitter norepinephrine, which affects many organs and causes the release of catecholamine hormones (epinephrine/norepinephrine and dopamine) from the adrenal medullae. Collectively, they initiate several physiological events to create the ‘fight-or-flight response’. Yet, the effects of weaning on catecholamines are equivocal, and their role in the regulation of the metabolism of the weaned pig is still unclear [55]. The HPA axis releases the corticotrophin-releasing hormone (CRH) during stressful events, which stimulates the release of the adrenocorticotropic hormone (ACTH) and glucocorticoids such as cortisol. Weaning results in a transient increase in CRH and cortisol levels, irrespective of weaning age, but they tend to be higher when weaned at a younger age [1,56,57]. Though activation of the HPA axis is a critical survival mechanism to respond to a stressor and return to homeostasis, e.g., by stimulation of gluconeogenesis and lipid metabolism, it is also to some extent responsible for compromised GIT structure and function after weaning.

At the intestinal level, the expression of CRH receptors is upregulated by early weaning [57] and chronic mixing and crowding [58]. The secretion of CRH from the hypothalamus and activation of intestinal CRH receptors cause hyperplasia and degranulation of mast cells in the intestinal mucosa [57,59,60]. Mediators of these mast cells, such as proteases, biogenic amines and cytokines, amongst others, are released, which negatively affect epithelial barrier function by influencing paracellular permeability [61,62]. For example, tryptase, the most abundant secretory granule-derived serine protease contained in mast cells, cleaves and activates a protease-activated receptor (PAR2) expressed on the apical and basolateral membranes of epithelial cells. This leads to Ca^2+^ mobilization, β-arrestin and ERK1/2 MAPK activity, perijunctional F-actin redistribution, zonula-1 delocalization and protein junctional adhesion molecule-A downregulation. Weaned piglets, therefore, may show a transient increased intestinal permeability to macromolecular markers lasting only a few days after weaning, which gradually declines over the subsequent 2 weeks [57,63]. Nevertheless, elevated CRH may also cause delayed gastric emptying, slow intestinal transit, increased colonic transit, and increased baseline secretory state and responsiveness to secretagogues of the intestine [62]. Regarding the latter, Moeser et al. (2007) [57] demonstrated that the neural inhibitor toxin, tetrodoxin, prevented weaning-induced elevations in pigs’ transepithelial short circuit current, a marker of intestinal ion secretion. Further, stress neuroendocrine mediators such as catecholamines and ACTH have also been shown to affect the adherence of *Escherichia coli* O157:H7, a serotype of Shiga toxigenic *E. coli*, to porcine cecal and colonic mucosa [64,65]. McCracken et al. (1999) [66] hypothesized that the increase in plasma cortisol may also inhibit the activity of AP-1 transcription factors, responsible for the induction of genes of the major histocompatibility complex class I (MHC-I) [67]. A compromised immune response, because of the reduction in MHC-I expression, may render pigs more susceptible to enteric infections, particularly viral infections, often resulting in diarrhea and reduced performance [66]. Moreover, in piglets aged 2–4 weeks, CD8+ T cells are largely absent in the lamina propria of the gut tissue [68]. Barrier dysfunction, secretory hypersensitivity and suppressed immunocompetence as a result of weaning stress may aggravate the post-weaning growth check and elevate susceptibility to enteric infections.

## 4. Socializing Piglets during Lactation in Commercial Settings

While pork industries typically cannot delay weaning for an excessively long period, techniques that can be adapted to a commercial setting to promote a more gradual weaning process have been explored and will be discussed below. Ultimately, pre-weaning socialization may reduce the risk of mixing stress and promote social eating behaviors (Section 4.1), while intermittent suckling (IS), a form of gradual weaning, can increase exploratory behaviors and allow the piglets to become accustomed to maternal separation (Section 4.2). Finally, a period of separation from the sow would also present an ideal window for piglets from different litters to co-mingle, allowing producers to combine the two techniques (Section 4.3).

### 4.1. Co-Mingling of Piglets before Weaning

#### 4.1.1. Co-Mingling as Management Strategy

The co-mingling (mixing) of non-littermate piglets before weaning is the subject of many studies and is of commercial interest due to its propensity to lower post-weaning stress responses. Kanaan et al. (2012) [69] described the co-mingling of non-littermates before weaning as a management strategy or process in which two or more unfamiliar litters can interact during the lactation period, allowing pre-weaning socialization and enabling piglets to cope better with future social and non-social challenges. Since it is common to socialize piglets that are similar in age, co-mingling is most applicable in batch-farrowing systems.

#### 4.1.2. Different Settings for Co-Mingling of Piglets before Weaning

Many variations of co-mingling exist, with most systems differing by individual versus group housing of the sow, the timing of when the piglets are co-mingled during lactation, the size of the co-mingled group, and the weaning age of the co-mingled piglets. Table 2 summarizes the different studies that exist according to these criteria and were all conducted in the last three decades.

Concerning the housing of the sow, the first approach of co-mingling non-littermates before weaning is applied in a conventional farrowing system (Table 2). Shutters in solid partitions between farrowing pens can be opened, solid barriers between farrowing pens can be removed, or solid partitions can be replaced by partitions with an opening a few days after farrowing, all allowing piglets from different litters to interact with each other. The sow remains confined in her farrowing crate. Either way, this intervention is relatively easy to implement in traditional farrowing facilities. Interestingly, Weary et al. (1999) [70] created a central communal area for the piglets by leaving two of the five farrowing pens vacant, whereas Turpin et al. (2017) [71,72] combined IS with co-mingling non-littermates by housing two different litters in an empty farrowing pen in an adjacent room. Another variant was described by Parratt et al. (2006) [73], where piglets from multiple litters had access to a common corridor after the removal of the separations along the corridor. Secondly, co-mingling of non-littermate piglets can also be implemented where sows are loose housed. Due to the absence of a farrowing crate or the removal of the farrowing crate shortly after partus, the sow can freely move in the farrowing pen. In a similar way as in conventional farrowing pens, the piglets can be co-mingled, yet the sow remains within her own pen. In the study of Kutzer et al. (2009) [74], for example, a loose-housing system with four farrowing pens was used. Thirdly, a multi-litter system, also referred to as a multi-suckling, group-housing or group lactation system, is a farrowing system in which a group of lactating sows is housed in one large pen together with their respective litters. More space, free movement for dams and progeny and social interactions between animals are the key elements of the system. Usually, a multi-litter system has a communal area, either divided or not into feeding, dunging and resting areas, and a number of farrowing pens. By removing partitions behind farrowing pens, removing the barriers and piglet hatches from the farrowing pens, removing the entire farrowing pen, or alternatively, rearranging the multi-litter area after farrowing, pigs obtain access to the entire area. A separate straw-bedded area for the piglets without access for the sows can be included [75,76]. In two studies, piglets were restricted from entering the communal area of the sows, but the piglets could move to other farrowing pens via a separate area that the sows could not enter [77,78]. Finally, a few days or weeks post-partum, sows and their litters could also be transferred from a conventional farrowing or loose-housing system to the multi-litter system.

Next to housing, the time of starting co-mingling can vary (Table 2). This moment is often expressed as a function of the age of the piglets at the start of co-mingling, even though the farrowing dates of the sows within the group may range over a few days. This makes the interpretation of the exact age of piglets at co-mingling less easy to define. Alternatively, reference is taken to the average farrowing date of the sows in the group, the average expected farrowing date of the sows, the farrowing date of the last sow in the experiment, or the day of weaning. In some cases, piglets are in the open-access area readily from birth [79]. Grouping of pigs from multiple litters can be performed early, i.e., 3 [80] or 5 days after farrowing [81,82]. In several studies, the co-mingling of non-littermates takes place around 10 days after partus, whereas fewer studies applied a co-mingling system where non-littermates are mixed from around day 14 after partus or beyond. Surprisingly, in the literature, minimal arguments are provided for the choice of the co-mingling date. Interestingly, in the study of Schrey et al. (2019) [83], non-littermates were allowed to freely choose the moment of socializing, which occurred on average at 10.6 days post-partum. Nonetheless, in most studies, piglets are grouped somewhere between 7 and 14 days after partus, which corresponds to the period of nest-leaving of the sow and her offspring and joining the herd, commensurate with increasing interactions with non-littermates in (semi-)natural pigs. Nevertheless, Parratt et al. (2006) [73] grouped unfamiliar piglets only 5 days before weaning, or approximately 16 days after birth. They hypothesized that the implementation of co-mingling of suckling piglets closer to weaning would have more benefits on pre-weaning performance and behavior since fewer suckling disruptions would occur. Unfortunately, few studies were designed to study the effect of time of co-mingling on aggression, stress, the occurrence of estrus during lactation, weaning-to-estrous intervals, sow performance, nursing–suckling interaction or piglet performance [84,85,86,87,88,89,90,91,92]. Yet, in practice, other considerations, such as vaccination schemes, disease prevalence, data collection from breeding sows and the provision of creep feed, may encourage later starting dates.

Another point of difference between studies is the size of the group, i.e., the number of litters (Table 2). In conventional farrowing systems or loose-housing systems, the number of grouped litters is usually small, i.e., non-littermates from two to four litters are grouped, whereas the multi-litter system mainly consists of four or more litters. The largest groups (8–22 sows and litters) were described by Hultén et al. [93,94,95,96].

**Table 2 animals-13-01644-t002:** Overview of different variants of pre-weaning co-mingling of non-littermates ^1^.

Reference	Housing of Sow in the Farrowing Unit		Time Start Co-Mingling of Suckling Piglets	Group Size and Composition	Weaning Age (Days)
	Individual/Group	Conventional Crates/Loose Housed			
Salazar et al., 2018 [84]	Individual	Conventional crates	7 days post-partum	2 litters	25
D’Eath, 2005 [97]	Individual	Conventional crates	10 days post-partum	2 litters	30 ± 3.0
Kanaan et al., 2012 [69]	Individual	Conventional crates	10 days post-partum	2 litters	18
Hong et al., 2017 [98]	Individual	Conventional crates	10 days post-partum	3 litters	/
Pluske and Williams, 1996 [42]	Individual	Conventional crates	10 days post-partum	3 litters	28.5
Kutzer et al., 2009 [74]	Individual	Conventional crates	10 days post-partum	4 litters	28
Morgan et al., 2014 [99]	Individual	Conventional crates	10 days post-partum	4 litters	21–24
Klein et al., 2016 [100]	Individual	Conventional crates	Youngest litter 10 days old	4 litters	28
Wattanakul et al., 1997 [101]	Individual	Conventional crates	11 days post-partum	3 litters	28
Weary et al., 1999 [70]	Individual	Conventional crates	11 days post-partum	3 litters	28
Hessel et al., 2006 [102]	Individual	Conventional crates	12 days post-partum	3 litters	28
Kanaan et al., 2008 [103]	Individual	Conventional crates	13 days post-partum	2 litters	/
Camerlink et al., 2018 [104]	Individual	Conventional crates	14 days post-partum	2 litters	26
Camerlink et al., 2019 [105]	Individual	Conventional crates	14 days post-partum	2 litters	26
Ko et al., 2020 [106]	Individual	Conventional crates	14 days post-partum	2 litters	25
Salazar et al., 2018 [84]	Individual	Conventional crates	14 days post-partum	2 litters	25
Weller et al., 2019 [107]	Individual	Conventional crates	14 days post-partum	2 litters	28
Parratt et al., 2006 [73]	Individual	Conventional crates	16 days post-partum (5 days before weaning)	2–3 litters	21
Van Kerschaver et al., 2021 [89]	Individual	Conventional crates	6, 11 and 16 days before weaning	3 litters	21
Turpin et al., 2017 [72]	Individual	Conventional crates	7 days before weaning	2 litters	22 ± 1.7
Turpin et al., 2017 [71]	Individual	Conventional crates	7 days before weaning	2 litters	25.3 ± 0.7
Illmann et al., 2007 [108]	Individual	Loose housed	10 days post-partum	2 litters	/
Kutzer et al., 2009 [74]	Individual	Loose housed	10 days post-partum	4 litters	28
Arey and Sancha, 1996 [79]	Group	Loose housed	At birth	4 sows and litters	/
Naya et al., 2019 [80]	Group	Loose housed	3 days post-partum	6 sows and litters	35
Bohnenkamp et al., 2013 [81]	Group	Loose housed	5 days post-partum	6 sows and litters	26
Bohnenkamp et al., 2013 [109]	Group	Loose housed	5 days post-partum	6 sows and litters	26
Bohnenkamp et al., 2013 [44]	Group	Loose housed	5 days post-partum	6 sows and litters	26
Grimberg-Henrici et al., 2016 [110]	Group	Loose housed	5 days post-partum	6 sows and litters	27.8 ± 0.15
Lühken et al., 2019 [111]	Group	Loose housed	Youngest litter 5 days old	6 sows and litters	26
Nicolaisen et al., 2019 [112]	Group	Loose housed	Youngest litter 5 days old	6 sows and litters	28
Nicolaisen et al., 2019 [113]	Group	Loose housed	Youngest litter 5 days old	6 sows and litters	28
Schnier et al., 2019 [82]	Group	Loose housed	Youngest litter 5 days old	6 sows and litters	26
Gentz et al., 2019 [114]	Group	Loose housed	5 days post-partum	10 sows and litters	27
Gentz et al., 2020 [115]	Group	Loose housed	5 days post-partum	10 sows and litters	27
Lange et al., 2020 [116]	Group	Loose housed	5 days post-partum	10 sows and litters	26.45 ± 0.97
van Nieuwamerongen et al., 2015 [117]	Group	Loose housed	Youngest litter 6 days old	5 sows and litters	27.1 ± 0.3
Grimberg-Henrici et al., 2018 [118]	Group	Loose housed	6 days post-partum	10 sows and litters	26 ± 1
Grimberg-Henrici et al., 2019 [119]	Group	Loose housed	6 days post-partum	10 sows and litters	26 ± 1
Verdon et al., 2019 [90]	Group	Loose housed	6.9 ± 1.2 days post-partum	5–7 sows and litters	26.7
Arellano et al., 1992 [120]	Group	Loose housed	7 days post-partum	5–10 sows and litters	28
Pedersen et al., 1998 [77]	Group	Loose housed	Youngest litter 7 days old	4 litters	28
Goetz and Troxler, 1995 [121]	Group	Loose housed	Youngest litter 7 days old	4 sows and litters	28
Rantzer et al., 1997 [122]	Group	Loose housed	7.2 (2–15) days post-partum	4 sows and litters	35
Verdon et al., 2020 [92]	Group	Loose housed	7.3 ± 1.2 days post-partum	5–7 sows and litters	25.5 ± 2.1
Verdon et al., 2019 [91]	Group	Loose housed	7.4 ± 1.1 days post-partum	5–7 sows and litters	25.5
van Nieuwamerongen et al., 2017 [123]	Group	Loose housed	7.9 ± 0.3 days post-partum	5 sows and litters	27.1 ± 0.4
van Nieuwamerongen et al., 2018 [124]	Group	Loose housed	7.9 ± 0.3 days post-partum	5 sows and litters	27.1 ± 0.4
van Nieuwamerongen et al., 2017 [125]	Group	Loose housed	8.1 ± 0.3 days post-partum	5 sows and litters	28 (A4 treatment) and 63 (IS9 treatment)
Maletínská and Špinka, 2001 [126]	Group	Loose housed	10 days post-partum	3–4 sows and litters	/
Kutzer et al., 2009 [74]	Group	Loose housed	10 days post-partum	4 sows and litters	28
Hillmann et al., 2003 [75]	Group	Loose housed	10 days post-partum	5 sows and litters	28
Marchant et al., 2000 [127]	Group	Loose housed	10 days post-partum	5 sows and litters	25.2
Schrey et al., 2019 [83]	Group	Loose housed	10.6 days post-partum	5 sows and litters	35
Verdon et al., 2020 [92]	Group	Loose housed	10.1 ± 1.2 days post-partum	5–7 sows and litters	25.5 ± 2.1
Li et al., 2010 [128]	Group	Loose housed	10 days post-partum	8 sows and litters	28–35
Li and Wang, 2011 [129]	Group	Loose housed	10 days post-partum	8 sows and litters	35
Hultén et al., 1995 [94]	Group	Loose housed	Youngest litter 10 days old	8–22 sows and litters	38.8
Hultén et al., 1998 [96]	Group	Loose housed	Youngest litter 10 days old	8–22 sows and litters	36.8 ± 6.9
Li and Johnston, 2009 [130]	Group	Loose housed	10 ± 3 days post-partum	8 sows and litters	28 ± 3
Dybjær et al., 2001 [131]	Group	Loose housed	11 days post-partum	6 sows and litters	32
Olsen et al., 1998 [132]	Group	Loose housed	11 days post-partum	6 sows and litters	32
Li et al., 2012 [133]	Group	Loose housed	12 ± 1.3 days post-partum	8 sows and litters	33 ± 1.3
Verdon et al., 2020 [92]	Group	Loose housed	13.5 ± 1.4–13.9 ± 0.9 days post-partum	5–7 sows and litters	25.5 ± 2.1
Verdon et al., 2019 [91]	Group	Loose housed	13.6 ± 1.3 days post-partum	5–7 sows and litters	25.5
Verdon et al., 2019 [90]	Group	Loose housed	13.9 ± 1.1 days post-partum	5–7 sows and litters	26.7
Weary et al., 2002 [78]	Group	Loose housed	14 days post-partum	3 litters	28
Wattanakul et al., 1998 [134]	Group	Loose housed	14 days post-partum	4 sows and litters	28
Verdon et al., 2016 [135]	Group	Loose housed	14 days post-partum	6 sows and litters	27.3
Tang et al., 2023 [85]	Group	Loose housed	8 and 13 days post-partum	5 sows and litters	49
Šilerová et al., 2006 [136]	Group	Loose housed	11–20 days post-partum (in one case, 7–10 days post-partum)	6–11 sows and litters	33–42
Šilerová et al., 2010 [137]	Group	Loose housed	11–20 days post-partum	6–11 sows and litters	33–42
Greenwood et al., 2019 [138]	Group	Loose housed	21 days post-partum	6 sows and litters	28
Thomsson et al., 2016 [87]	Group	Loose housed	1–3 weeks post-partum	2–6 sows and litters	44 ± 1.6
Thomsson et al., 2015 [86]	Group	Loose housed	1–3 weeks post-partum	5–8 sows and litters	42
Thomsson et al., 2018 [88]	Group	Loose housed	1–3 weeks post-partum	5–8 sows and litters	43.6 ± 0.6–44.2 ± 0.6
Hultén et al., 1995 [93]	Group	Loose housed	Youngest litter 2 weeks old	11–22 sows and litters	37.7
Wattanakul et al., 1997 [76]	Group	Loose housed	2 weeks post-partum	5–6 sows and litters	31 (trial 1) and 29 (trial 2)
Hultén et al., 1997 [95]	Group	Loose housed	2–3 weeks post-partum	12–22 sows and litters	39.2

^1^ References are ordered according to housing of the sows in the farrowing unit, time of start of co-mingling of suckling piglets, group size and composition.

#### 4.1.3. Occurrence of Cross-Suckling or Allo-Suckling in Piglets by Co-Mingling before Weaning

Co-mingling of non-littermates before weaning gives piglets the opportunity to suckle sows other than their mother [126]. Lower milk production is the main reason that piglets become ‘cross-sucklers’ [132], and cross-sucklers are able to search for sows with higher milk production (as assessed by piglet weight gain) and to have a preference for more productive teats [132]. This may be beneficial for their growth [70], but it has also been reported to have no effect on performance when compared to non-cross-sucklers [108,132]. More specifically, piglets that cross-suckled in more than 50% of their nursings (i.e., permanent allo-sucklers) were found to have an average weight gain of approximately 2300 g, whereas their biological littermates that were not classified as cross-sucklers had an average weight gain of approximately 2900 g between day 10 post-partum (the day of co-mingling) and day 24 post-partum. Dybjaer et al. (2001) [131] observed that the daily growth of piglets was negatively correlated with the percentage of cross-sucklers. Indeed, overt cross-suckling may cause lost nursings [77] and more disturbances of the orderly suckling and continuation of teat order [70]. In some studies, the incidence of cross-suckling appeared to be very low [81,101], while in the experiment of Maletínská and Špinka (2001) [126], more than one-third of the piglets would cross-suckle at least once. In some studies, the incidence of cross-suckling was even higher, e.g., more than 50% [76,134].

This high variability in cross-sucking behavior is surprising. The occurrence of cross-suckling is affected by several factors [126,131] and may depend on the manner of implementation of the system of co-mingling non-littermates [78,113,134]. For example, the incidence of cross-suckling may decrease when sows and their respective litters have the opportunity to establish clear dam–progeny bonds prior to implementing co-mingling [78,113,134]. Due to intense sow–piglet interactions immediately after initiating the co-mingling of non-littermates, sows seem to be highly discriminant against other, alien piglets (potential cross-sucklers), which leads to a reduced number of successful sucklings. However, 3 h after grouping, cross-sucklers seem to be accepted by the sow [101], which is reasonable since it is hypothesized that piglets pick up the same odor after grouping. Horrell and Hodgson (1992) [139] observed that sows cannot distinguish their own piglets from other piglets until the age of 2 weeks, when the odors are masked. In contrast, piglets seem to be less able to distinguish their littermates from non-littermates [140], suggesting they may unintentionally join non-littermates at the udder of their sow during milk letdown.

#### 4.1.4. Effects of Co-Mingling before Weaning on Social Behavior before and after Weaning

The behavioral effects of pre-weaning socialization have been the focus of many recent studies, with observations of the behaviors of socialized piglets both before and after weaning. When piglets are first socialized with non-littermates during lactation, aggressive behaviors, reflected in the occurrence of skin lesions or damage, are often observed immediately after litters are mixed [89]. This type of fighting usually occurs between unfamiliar pigs and is most likely related to the establishment of a social hierarchy. This type of competitive behavior between non-littermates is also evident at the udder when cross-suckling occurs. Interestingly, Salazar et al. (2018) [84] observed an increase in aggression 1 day after mixing in piglets co-mingled at 7 days of age, which was not the case when piglets were co-mingled at 14 days of age. In contrast, Verdon et al. (2020) [92] found no effect of age on co-mingling and suggested that this discrepancy could be related to differences in the studied systems. In a study by Kanaan et al. (2008) [103], co-mingled piglets had more ear injuries 2 days after socialization compared with conventionally housed piglets, but the differences disappeared by day 5 after socialization. Kutzer et al. (2009) [74] found no effect of the housing system on skin lesion scores 4 days after allowing unfamiliar litters to co-mingle during lactation. Only on the last day in the farrowing pens did the grouped piglets have more skin lesions. Furthermore, in some studies, no differences in lesion scores between socialized and non-socialized piglets were observed (anymore) close to weaning [92,106,135] or at any timepoint [69]. The size and layout of the pens might explain why the co-mingled piglets had more aggression than others across the different studies [74]. In this respect, the size of the co-mingled group also seems to be a factor for aggression. Grimberg-Henrici et al. (2018) [118] reported more skin lesions on the carpus and body of co-mingled piglets of group-housed sows at the end of the lactation period compared with conventionally reared piglets. They indicated that an increase in social interactions due to a large (*n* = 10 litters) group size of co-mingled litters might be the reason since, in the study by van Nieuwamerongen et al. (2015) [117], where only five litters were co-mingled before weaning, no differences in skin lesions on the body were found at the end of the lactation period. However, in contrast to the study of Grimberg-Henrici et al. (2018) [118], the multi-suckling piglets in the study of van Nieuwamerongen et al. (2015) [117] had more skin lesions at the snout compared to the control piglets at that time, which, according to the authors, was due to the increased competitive behaviors of the piglets at the udder and, in consequence, the incidence of cross-suckling.

It is after weaning that the behavioral benefits of co-mingling are observed. Hillmann et al. (2003) [75] demonstrated that piglets that socialized with non-littermates before weaning adapted better to social and non-social challenges during the weaning event. In addition, numerous studies have demonstrated a reduced level of aggression among socialized piglets immediately after weaning, independent of the housing system, group size and age at the start of co-mingling. For example, Van Kerschaver et al. (2021) [89] showed that co-mingled piglets had fewer skin lesions at the shoulders and the flanks compared to controls in the days following weaning. Socialized piglets are already familiar with one another as, prior to weaning, the social skills of the piglets were stimulated and the piglets had already established a new social hierarchy. Some studies demonstrate that co-mingling prior to weaning reduces pig aggression at weaning even when familiarity at weaning is not a factor, i.e., when piglets that are mixed at weaning originate from different co-mingled groups. In addition, pre-weaning familiarity may also have benefits in the longer term for social development, as evidenced by, for instance, less fighting [129], a shorter duration of fights [97,129], fewer skin lesions [104,105] and fewer shorter tails caused by tail-biting in the fattening period [100], even when socialized piglets are regrouped with unfamiliar piglets at a later stage during rearing. However, other studies report that aggression after weaning did not differ between socialized and control pigs [99,137]. Additionally, Naya et al. (2019) [80] and Gentz et al. (2020) [115] found no clear effect on tail biting in the early social environment occurring before weaning. Differences in the group size, a strong batch effect, inter-farm differences, weaning age (and moment of observations), method of weaning (without mixing litters), stress and space allowance and the method of assessing social behaviors may explain different outcomes across studies. Indeed, it is important to understand that, in the literature, various methods and techniques, such as (skin) lesion scoring, social (challenge) tests such as resident–intruder tests, video recordings and behavioral observations, are used to assess the social skills of co-mingled piglets before and after weaning.

#### 4.1.5. Effects of Co-Mingling before Weaning on Performance before and after Weaning

The main objective of the co-mingling of non-littermates prior to weaning is to ameliorate the weaning transition in order to reduce GIT dysbiosis and maintain optimal production. The co-mingling of non-littermates does not commonly affect the performance of piglets before weaning [97,104,138], although some studies have reported lower weaning weights and pre-weaning weight gains in socialized piglets compared with non-socialized piglets [44,76,92,127]. According to the authors, this could be due to a myriad of factors, including: the timing of co-mingling; higher space allowances and, therefore, more exercise; lower creep feed intake; disruption of suckling; cross-suckling; enhanced active behaviors; greater occurrence of fighting behaviors (skin lesions); relocation of the sows during co-mingling within the co-mingling group and, consequently, a high incidence of cross-suckling; and/or stress on the piglets due to the introduction of the piglets into new accommodation and environments after moving the sows and their litters from conventional housing to group lactation. In contrast, Arey and Sancha (1996) [79] observed a higher piglet daily weight gain in week 2 post-partum in piglets born in a multi-suckling system and suggested that this was due to the more successful nursing behavior in the multi-suckling system.

Several studies have demonstrated a positive influence on the post-weaning performance of piglets, particularly in the immediate post-weaning period, suggesting that co-mingling might reduce weaning stress and contribute to a better adaptation of the piglets to weaning. For example, in the study of Hessel et al. (2006) [102], piglets co-mingled in a conventional housing system tended to gain 290 g more after weaning than conventionally reared piglets during the first week after weaning, and gained, in total, 1.09 kg more over the entire study (9 weeks). Feed efficiency in the first week after weaning tended to be improved in piglets that were co-mingled 6 days before weaning [89]. In research by van Nieuwamerongen et al. (2015) [117], piglets reared in a multi-litter system gained 69% more weight between day 1 before weaning and day 2 after weaning compared with control piglets, and the weight gain of these piglets was 24% higher in the total nursery period, i.e., between days 1 and 35 after weaning. The piglets from the multi-litter system also consumed 81% more feed between days 1 and 2 after weaning compared with the conventionally reared piglets. In the study of Weary et al. (2002) [78], co-mingled piglets consumed more creep feed before weaning and one day after weaning, resulting in higher weight gain in the immediate post-weaning period. Co-mingling non-littermates before weaning may thus also positively affect feed intake in pigs. However, these substantial benefits have been contradicted by others. Other studies showed that co-mingling did not affect post-weaning performance [42,109]. Although the body weight was higher at weaning in the co-mingled piglets, the average daily gain during the 14 days after weaning did not differ between the piglets in the co-mingled group and in the conventionally reared group in the study by Pluske and Williams (1996) [42] (367 vs. 378 g/day). Bohnenkamp et al. (2013) [109] observed no differences in body weight at weaning and after rearing (7 weeks after weaning) between co-mingled piglets and control piglets and suggested, by comparing other studies, that the timing of grouping non-littermates during lactation might have had an impact. Rantzer et al. (1997) [122] observed an even lower growth rate during the 4 weeks after weaning of piglets from a multi-suckling system compared with conventionally reared piglets, which the authors ascribed to the larger weaner pen and the lower feed intake following weaning of the piglets from the multi-suckling system.

#### 4.1.6. Effects of Co-Mingling before Weaning on Mortality before and after Weaning

Mortality and, more specifically, pre-weaning mortality of piglets may depend on the housing system used to achieve co-mingling. Pre-weaning piglet mortality is typically higher in multi-litter systems (group lactation) and is often caused by a greater incidence of crushing. For example, in the study of Marchant et al. (2000) [127], 14–17% of the piglets born alive died in the multi-suckling systems as a result of crushing, compared to 8% in the conventional farrowing system. In addition, van Nieuwamerongen et al. (2015) [117] found that more piglets died before weaning in the multi-litter system compared with the conventional farrowing system (respectively, 3.22 ± 0.42 vs. 1.52 ± 0.25 piglets per litter), particularly because of crushing (respectively, 2.34 ± 0.44 vs. 0.20 ± 0.09), which is likely associated with the loose farrowing pens in the multi-litter system since mortality mostly occurred before co-mingling. Interestingly, in some studies, no differences in piglet losses during lactation between a multi-suckling system and a single housing system were observed, which was likely due to housing sows and their litters individually for a short period following birth before transferring to a multi-suckling system [76,87,135], the presence of farrowing crates in the multi-suckling system [44] or the enhanced maternal behavior of the sows in an enriched environment [79]. In the study of Grimberg-Henrici et al. (2016) [110], piglet losses before weaning were even lower in the group-housing system compared with conventional farrowing crates. The researchers suggested that the experience of the stockpersons and the free movement of the sows before farrowing explained this. The mortality rate of piglets in conventional housing systems where co-mingling of non-littermates is applied seems to be lower and does not differ from conventional housing [97,102,104]. After weaning, no significant differences in piglet mortality between co-mingling and conventional housing systems have been found.

#### 4.1.7. Effects of Co-Mingling before Weaning on GIT Structure and Function and Immunocompetence of Weaned Piglets

Co-mingling of non-littermates before weaning may also positively influence the GIT structure and function and immunocompetence of weaned piglets. In a study by van Nieuwamerongen et al. (2015) [117], the average fecal consistency score was significantly lower (i.e., more solid feces and less diarrhea) during the 2 weeks after weaning for the piglets reared in the multi-litter system compared with the conventionally reared piglets. In addition, the study showed that titers of IgM-binding keyhole limpet hemocyanin, as a measure for natural antibodies of the adaptive immune system, were higher for group-housed suckling piglets compared with conventionally housed piglets on the day of weaning. They speculated that this was due to an accelerated development of the pigs’ immune system, triggered by early contact with materials such as feed and/or enrichment in the pre-weaning environment. However, titers of IgG-binding keyhole limpet hemocyanin and haptoglobin and leukocyte concentrations were not affected by the housing system. In the experiment by Rantzer et al. (1997) [122], in which a multi-suckling system was compared with a conventional housing system, the average diarrhea score did not differ after weaning, despite the fact that the peak excretion of hemolytic *E. coli* was delayed in piglets from the multi-suckling system. The fecal consistency scores after weaning also did not differ between pigs reared in a multi-litter system and a conventional system in the study of van Nieuwamerongen et al. (2018) [124]. However, in the study, lower plasma sugar concentrations such as mannitol and galactose were observed on day 5 after weaning in pigs raised in the multi-litter system. The authors speculated that the lower concentrations of mannitol would probably indicate a lower intestinal permeability and, therefore, a less compromised intestinal barrier function as a result of weaning. Further, it was speculated by the authors that the lower concentrations of galactose were due to a better adaptation to weaning and the pre-weaning experience with solid feed of the pigs in the multi-litter system since no differences in post-weaning feed intake were observed.

### 4.2. Gradual Separation from the Sow

#### 4.2.1. Sow-Controlled Housing to Promote Gradual Weaning

To overcome the negative consequences of abrupt weaning, housing systems that allow the sow to leave her piglets voluntarily have been investigated. In sow-controlled housing, sows can express their natural tendency to spend time apart from their offspring as lactation advances by leaving the conventional farrowing pens to mingle with other sows and/or eat in a common area, whereas the piglets cannot leave the farrowing pen. In these types of systems, sows will generally mimic the pattern of spending less time with their litters during the latter part of lactation [26,141,142]. However, there is still a considerable amount of variation in the response patterns, with some sows either spending all their time away from the litter or those that rarely leave the nest [143]. Further to this, no consistency from one lactation to another has been seen [143]. Compared with animals in conventional systems, piglets from sow-controlled housing generally consume more solid food during lactation [141,144], which can translate into an increase in solid feed intake and growth rate after weaning [144,145]. However, there is a risk that the large variation in the use of piglet-free areas may lead to inadequate maternal care, and it has therefore been suggested that housing systems that allow for more consistent use of piglet-free areas may be more beneficial [144].

#### 4.2.2. Intermittent Suckling to Promote Gradual Weaning

Similar in concept to sow-controlled housing, IS is a more structured, gradual weaning process that involves the daily separation of sows and piglets for a specified period during the last part of lactation. There are two main reasons why IS has been examined over the years. The first reason is to induce estrus in lactation, allowing for mating in lactation, which presents an opportunity to uncouple estrus cycling from weaning, therefore extending the weaning age (e.g., to 5–8 weeks) without decreasing the number of litters produced per sow per year ([146,147]; for reviews). Second, IS has also been studied as a strategy to promote supplementary (e.g., creep) feed intake in piglets during lactation. In this scenario, conventional weaning ages (i.e., 3 to 4 weeks) have been used in combination with intermittent separation of the litter from the sow to promote exploratory behaviors and encourage piglets to explore nutritional options other than milk. Table 3 describes the different variations in IS regimens that have been examined since 1961.

#### 4.2.3. Intermittent Suckling and Extended Lactation: Effects on Litter Performance

Weaning under current commercial conditions can have a negative impact on production, health and welfare in the immediate post-weaning period. An older weaning age (i.e., greater than 5 weeks) has been shown to improve post-weaning feed intake along with a reduction in stress [148] and a reduction in post-weaning diarrhea [149]. However, given that sows are naturally in anestrus during lactation, producers rely on shortening the length of lactation to maximize the number of litters born per sow per year, thereby creating a conflict between sow and piglet welfare and the profitability of sow performance [150]. Previous studies have reported that 10–16% of commercial sows housed individually will ovulate spontaneously during lactation [151]. This can be increased if interventions, such as group lactation housing, boar exposure and/or exogenous gonadotropins, are introduced [151,152]. Reducing the suckling stimulus in a controlled manner (i.e., not relying on sows to regulate nursing frequency) by subjecting litters to interrupted suckling techniques such as IS has been used on its own or in combination with the techniques mentioned above to increase gonadotrophin release and can result in a 100% occurrence of estrus in lactation. However, the results are still highly variable, most likely due to factors such as sow parity, metabolism, litter suckling intensity, seasonal effects, and litter creep feed intake [147,150].

While the effects of IS on sow reproductive performance are well documented, less is known about the impacts that IS in combination with extended lactation can have on piglet performance. A reduction in weight at the start of the intervention followed by compensatory growth a week later, most likely due to an increase in creep feed intake, is consistent across most of the early studies [153,154,155]. The effects of IS in combination with extended lactation were compared with control groups weaned at a conventional weaning age in a series of studies. In summary, combining IS with an extended lactation period caused a more gradual adaptation to weaning with respect to piglet growth and feed intake, which significantly ameliorated the post-weaning growth check [156,157,158]. However, a mild growth check was evident at the start of IS when IS started at 19 days of age or less [156,157]. This growth check was not evident when IS started at 26 days of age or older [157,158]. Collectively, these data suggest that, when IS starts at an older age, creep feed intake is likely to be sufficient to compensate for the reduction in milk consumption during the time of separation from the sow.

Studies examining the effect of extended lactation on GIT morphology before weaning are limited, with only Turpin et al. (2016) [158] reporting that IS, with or without an extended lactation, did not affect the absorption of mannitol, a measure of absorptive surface area, compared with conventionally weaned controls 3 days before weaning (i.e., 4 days after the start of IS). After weaning, IS in combination with an extended lactation (33 days) prevented villus atrophy compared with conventionally weaned piglets [157]. Results from Nabuurs et al. (1996) [159] supported this finding and suggest that creep feed intake before weaning is an important factor in this outcome. Four days after weaning, Turpin et al. (2016) [158] reported an improvement in galactose absorption but no improvement in mannitol absorption, despite a marked increase in solid feed intake before and after weaning in litters subjected to IS with an extended lactation compared with conventionally weaned pigs. The potential limitation of mannitol as a measure of the intestinal surface area needs to be considered when interpreting these results [158], but the reported increase in galactose absorption highlights the importance of solid feed intake in GIT function, given that a change from a milk-based diet to solid feed increases the Na^+^-K^+^ ATP pump activity needed for galactose absorption via a sodium–glucose-linked transporter [160].

**Table 3 animals-13-01644-t003:** Overview of different variants of intermittent suckling (IS).

Reference	Start Time of Gradual Weaning ^1^	Separation Length ^2^	Duration	Weaning Age	Combined with Another Technique
Smith, 1961 Experiment 1 [153]	Days 31–35	12 h per day	25–21 days	56 days	Extended lactation + mating in lactation
Smith, 1961 Experiment 2 [153]	Day 21	12 h per day	35 days	56 days	Mating in lactation + extended lactation
Kirkwood and Smith, 1983 [161]	Day 14	12 h per day	14 days	28 days	Mating in lactation
Henderson and Hughes, 1984 [154]	Day 10	12 h per day	25 days	35 days	Mating in lactation + extended lactation
Grinwich and McKay, 1985 [155]	Day 21	3 or 22 h per day	14 days	35 days	Mating in lactation + extended lactation
Chapple et al., 1989 [162]	Day 14	4 × 5 h intervals per day with 1 h back with the sow in between	14 days	28 days	+/− hydrocortisone injections
Costa and Varley, 1995 [163]	Day 12	3 h per day	9 days	21 days	Mating in lactation
Nabuurs et al., 1993 [164] Experiment 3 only	Days 16–18	8 h per day	14 days	30–32 days	Control piglets not offered creep feed
Nabuurs et al., 1996 [159]	Day 18	8 h per day	14 days	32 days	With and without creep feed (experiment 2) Control piglets did not receive creep feed
Kuller et al., 2004 [165]	Day 16	12 h per day	11 days	27 days	
Kuller et al., 2007 [166]	Day 14	12 h per day	11 days	25 days	
Berkeveld et al., 2007 [156]	Day 14	12 h per day and 2 × 6 h intervals per day with 6 h back with the sow in between	27 to 31 days	41 to 45 days (controls were 21 days)	Extended lactation
Millet et al., 2008 [167]	Day 14	7 h per day	14 days	28 days	Flavor recognition
Berkeveld et al., 2009 [157]	Day 19 or 26	10 h per day	7 or 14 days	26 or 33 days	Extended lactation for two of the four treatment groups
Kluivers-Poodt et al., 2010 [168]	Day 14	12 h per day and 2 × 6 h intervals per day with 6 h back with the sow in between	Not specified	23 days after ovulation	Measurements in sows only
Downing et al., 2011 [169]	Day 14, 16 or 18	16 h per day	3 days	26 days	Mating in lactation
McDonald et al., 2013 [170]	Day 21	16 or 8 h per day	3 days	28 days	Mating in lactation
Frobose et al., 2015 [171]	Day 18	12 h per day	7 days	25 days	Co-mingling and alternating between two sows every 12 h, split weaning, 24 h separation
Turpin et al., 2016 [158]	Day 21 or Day 28	8 h per day	7 days	28 days or 35 days	Extended lactation for one of the three treatment groups
Turpin et al., 2016 [172]	Day 22	8 h per day	7 days	29 days	
Turpin et al., 2017 [71]	Day 18	8 h per day	7 days	25 days	One of the three treatment groups exposed to co-mingling
Turpin et al., 2017 [72]	Day 15	8 h	7 days	22 days	2 × 2 factorial with IS and co-mingling as main effects
Turpin, 2017 [173]	Day 20 or 23	8 or 16 h	6 days or 3 nights	26 days	
van Nieuwamerongen et al., 2017 [125]	Day 35	10 h	7 days	63 days	Multi-suckling, sow-controlled housing, mating in lactation and extended lactation

^1^ Expressed as day of lactation. ^2^ Pattern continued from start time to weaning unless otherwise specified.

#### 4.2.4. Intermittent Suckling and Conventional Lactation: Effects on Litter Performance

Many studies have combined conventional weaning ages (i.e., 3 to 4 weeks) with an IS regimen to improve creep feed intake by promoting exploratory behavior when the piglets are separated from the sow. However, the length of time that the piglets are separated from the sow, the style of separation housing and the age at which IS is initiated all seem to play a role in the outcomes. Litters subjected to separation times of 12 h for 7 days consistently had an increase in creep feed intake compared with control litters that remained with the sow [166,166,171], and this translated into improvements in solid feed intake after weaning [165,166]. Non-significant increases in creep feed intake have also been reported with 12 h separation times [154]. There are a limited number of studies examining separation times greater than 12 h. Chapple et al. (1989) [162] used a total separation time of 20 h per day for 14 days and did not find a difference in creep feed intake between limited nursing piglets and controls, but the piglets did return to the sow every 5 h for 1 h. While the authors postulated that this could have been due to the acceptability of the diet or litter-to-litter variability of feed intake (discussed below), this result supports findings from Berkeveld et al. (2007) [156], where segmenting the separation time over multiple periods in a day resulted in lower feed intake levels during lactation, probably due to a greater dependency on milk. Turpin (2017) [173] examined separation times of 16 h over 3 days and found an increase in creep feed intake, although these results need to be interpreted with some caution because creep feed was already stimulated in the treatment group before the start of the IS intervention, making it difficult to conclude that IS was the only contributing factor for these piglets.

When separation times were less than 12 h, the ability of IS to stimulate creep feed intake was not guaranteed, with IS litters either consuming more [71,72,158] or the same amount of creep feed as the control litters that remained with the sow [71,157,158,172,174]. Only one study reported a reduction in feed intake in IS litters compared with continuously suckled litters [167]. Collectively, these outcomes suggest that IS litters are more likely to remain dependent on milk for their growth when separation times are less than 12 h. One of the challenges to consider when assessing strategies to improve creep feed intake is that individual creep feed consumption is highly variable [33,34,35]. Studies that have looked at the individual eating patterns of piglets exposed to IS have reported that IS increases the creep intake of piglets that were already eating before the period of separation but does not increase the percentage of piglets eating creep feed within a litter [166,173]. However, when litters were categorized as either high or low creep feed intake litters, the piglets in the low creep feed intake group that were subjected to IS visited the feeder more frequently than their control counterparts, which translated to more time spent at the feeder on the second day after weaning [175].

In conventional production systems, increases in creep feed consumption can be achieved as lactation progresses, with studies reporting 60–80% of total creep feed intake occurring in the 6 days before weaning [7,176] and higher feed intake rates achieved (in a non-linear pattern) as the piglets’ age increases [177]. This phenomenon also occurs with gradual weaning strategies, where postponing the onset of IS for 1 week (14 days vs. 21 days) caused a greater increase in creep feed intake [178]. In addition to the length of the daily separation period and the age at which separation from the sow starts, the style of separation housing has not received much attention in studies to date, but housing IS litters in the same room as their control counterparts will likely have a negative impact on creep feed intake since nursings are usually synchronized within a room and are accompanied by noise, which could distract the IS litters [165].

Most IS studies using conventional weaning ages with IS starting in the week before weaning have observed a reduction in piglet growth rate at the start of IS when the separation period is greater than 7 h per day [72,157,158,162,165,166,170,173], with the exception of only a couple of studies where IS had no impact on growth rates before weaning compared with continuously suckled litters [71,172]. In most cases, piglets compensated for the reduction in growth at the start of IS during the latter part of the first week after weaning [72,156,163,166,172], but in contrast to studies where IS is combined with extended lactation, it seems that the weaning-associated growth check in the first few days after weaning is not necessarily prevented or reduced when IS for one week or less is implemented with a conventional weaning age [71,72,157]. Furthermore, there has been no evidence to suggest that gradual weaning through IS improves longer-term growth or market weights [158,166,171].

#### 4.2.5. Intermittent Suckling and Conventional Lactation: Impacts on the GIT, Behavior and Welfare

Despite varying results regarding pre-weaning creep feed intake and growth, no studies have reported gradually weaned pigs performing worse in the immediate post-weaning period compared with pigs subjected to abrupt weaning. This suggests that the beneficial effects of gradual weaning might also be mediated by factors other than the potential for improvements in feed intake. This is supported by findings from Berkeveld et al. (2009) [157], where one week of IS (10 h of separation from the sow per day) before weaning at day 26 prevented post-weaning villus atrophy, even though feed intake before and within the first 2 days after weaning was not improved compared with conventionally weaned pigs. To date, only a few studies have examined the behavior of gradually weaned pigs after weaning. More focus has been placed on piglet behavior before weaning and during IS because of the perceived welfare implications of intermittent sow and piglet separation and udder damage from excessive piglet attention after re-joining with the sow. Results from continuous and instantaneous scan sampling studies have reported that IS pigs showed the lowest levels of manipulative behavior [71,174] and the highest levels of relaxed behavior [71] on the day of or the day after weaning compared with abruptly weaned pigs. These differences in behavior were only supported by reductions in the stress marker cortisol when the blood samples were taken on the day of or 24 h after weaning [71,72,174].

The act of repeated maternal separation associated with gradual weaning during lactation on the piglets’ stress response is a potential welfare concern [179]. Berkeveld et al. (2007) [156] reported a peak in total piglet activity and vocalization on the first day of IS during the 12 h separation, but 2 days later, the total piglet activity had decreased and then stabilized for the remainder of lactation. This result was supported by the study of Turpin et al. (2016) [172], in which there was an elevation in plasma cortisol after the first 4 to 6 h of separation for the IS piglets compared with the continuously suckled piglets, but this effect disappeared by the next measurement point 6 days later. A similar pattern was observed in sow cortisol levels during the first separation period from the piglets using 12 h separation times [168]. Piglets subjected to long periods of separation (16 h overnight for 5 nights at 2 to 3 weeks of age) had an increased probability of passive lying behavior during separation times compared with continuously suckled controls, which can be interpreted as a stress behavior [38]. This suggests that extended separation times could have a detrimental effect on piglet welfare [180]. In addition to cortisol, other markers of piglet well-being, including acute phase proteins and cytokines, have been measured before and after weaning in piglets subject to IS across a number of studies using mostly an 8 h separation for 6 to 7 days, but with no obvious patterns in response [71,72,157,172,173].

### 4.3. Gradual Weaning in Combination with Co-Mingling

Group lactation systems that promote piglets spending gradually less time with their sows in combination with pre-weaning socialization generally result in improvements in post-weaning behavior and performance compared to conventionally weaned pigs. Similar results have also been achieved when gradual weaning and piglet socialization are applied in a more structured way. Turpin et al. (2017) [71] investigated the behavior and performance of litters that were separated during IS treatments (8 h per day for 7 days) and then socialized with another litter during the separation period. The addition of socialization with another litter before weaning improved pre- and post-weaning feed intakes and post-weaning growth rates in an additive manner when the socialized groups could remain together after weaning. This result highlights the advantage of familiarity within a group rather than the development of social skills since similar findings were not reported in other studies where co-mingling was combined with IS using a conventional weaning age (i.e., 3 to 4 weeks) [71,171]. Applying co-mingling intermittently may have had an impact on these outcomes, and the requirement for extra labor and potentially specialized housing to achieve IS in combination with co-mingling may limit the commercial application of these techniques.

Nevertheless, there have been promising performance, health and behavioral results from a study that has combined all the techniques described in this review: multi-suckling, IS, sow-controlled housing, and extended lactation [125]. In this study, all piglets were subjected to a multi-suckling system with five sows in each group. One treatment group was abruptly weaned at 4 weeks of age, and the other was subject to IS, involving 10 h of separation per day during the fifth week of lactation. After this, the sows were then able to separate themselves from the piglets voluntarily until weaning at 9 weeks of age. The authors chose to compare results by age, but it must be recognized that both the duration of the suckling period (4 weeks vs. 9 weeks) and the physiological age will have had an influence on the measured parameters. Nonetheless, the combination of multi-suckling with IS and extended lactation prevented the growth check at the start of IS and at weaning [125]. The multi-suckled IS pigs also showed a lower level of belly-nosing and a more solid fecal consistency, indicating a lower occurrence of diarrhea, compared with the piglets that were only exposed to multi-suckling and weaned at 4 weeks of age. Regarding reproductive performance, 83% of the treatment sows showed an estrus in lactation, and of those sows, 95% were pregnant following insemination during lactation [125]. These results highlight that some adjustments to current multi-suckling housing designs can facilitate the implementation of IS and/or sow-controlled housing; therefore, removing the need to wean early without sacrificing sow performance.

## 5. Conclusions

It was hypothesized in this review that pre-weaning socialization presents an opportunity to reduce the risk of mixing stress and promote social eating behavior, while intermittent suckling (IS), a form of gradual weaning, can increase exploratory behavior and allow the piglets to become accustomed to maternal separation. Numerous studies demonstrated a reduced level of aggression among socialized piglets immediately after weaning, independent of the housing system, group size and age at the start of co-mingling prior to weaning. Studies have generally shown a positive influence on post-weaning performance, particularly in the immediate post-weaning period. Yet, attention is needed to prevent reductions in pre-weaning weight gains due to the timing of co-mingling, a higher space allowance and, therefore, more exercise, a lower creep feed intake, the disruption of suckling, cross-suckling, enhanced active behaviors and others. In the majority of publications, piglets from different litters were co-mingled between 7 and 14 days of age, which seems to be consistent with their (semi-)natural behavior. The size of the group seemed to depend on the housing system; in multi-litter systems, groups consisted mainly of four or more litters, whereas, in other systems, it was usually restricted to four litters or less. It is not clear whether this is due to practical limitations or scientific grounds, yet larger groups may cause more aggression between piglets and competitive behavior at the udder. Overall, few studies have consistently studied main variables, such as housing system, group size and age at the start of co-mingling prior to weaning, which prevents us from drawing firm conclusions on the effect of these factors.

From the piglet perspective, although there are well-known impacts of IS on sows’ reproduction during lactation, IS has also been studied as a strategy to promote creep feed intake during lactation. Many studies have combined conventional weaning ages (i.e., 3 to 4 weeks) with an IS regimen to improve creep feed intake by promoting exploratory behavior when the piglets are separated from the sow. However, the length of time and frequency with which the piglets are separated from the sow, the style of separation housing and the age at which IS is initiated all seem to play a role in the outcomes. Intermittently suckled litters are more likely to remain dependent on milk for their growth when separation times are less than 12 h per day. Nonetheless, no studies have reported gradually weaned pigs performing worse in the immediate post-weaning period compared with pigs subjected to abrupt weaning. This suggests that the beneficial effects of IS might also be mediated by factors other than the potential for improvements in feed intake. If IS is applied with an extended lactation period, creep feed intake is likely to be sufficient to compensate for the reduction in milk consumption during the time of separation from the sow.

Pre-weaning socialization, or co-mingling, and IS are techniques that are potentially adaptable to a commercial setting to promote a more gradual weaning process and reduce post-weaning stress for piglets with better welfare outcomes and more optimal performance and GIT structure and functioning. The economic benefits of either or both practices, if implemented commercially, will be determined by the level of performance and/or welfare improvements and impacts on other variables, for example, a reduction in medication use after weaning.

## Data Availability

Data sharing not applicable.

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
