# Peer review of "Reducing Weaning Stress in Piglets by Pre-Weaning Socialization and Gradual Separation from the Sow: A Review"

_animals, 2023, doi:10.3390/ani13101644_

Round 1

Reviewer 1 Report

The paper is interesting and relevant to the current welfare problem facing weaning of piglets and their welfare and the welfare of the sow. Interestingly the authors have used articles published during the 1980s to show the importance of group pre-weaning in piglet welfare and the benefits at the time of weaning. According to the EU directive 2008/120/EEC, the minimum age of weaning is at 28 days of age. Therefore, the group pre-weaning methods as were followed in the 80s and 90s with modern technology could benefit the commercial production of pigs.

The authors have reviewed recent publications, but it would be necessary throughout the article, at certain important points compare the methods previously used with those in current use.

The paper is following the American English system. The suggestion is for the authors to use the English that is used in Europe as the publication is in a European journal.

  Simple Summary: Please verify

 In commercial pork production piglets are weaned at a young age, i.e., mostly between 2.5 and 4.5 weeks of age.

1. Introduction: No comments

2. Development of social and eating behavior of pigs in (semi)-natural conditions

                2.1. Nest occupation and maternal care

                Lines: 86-89. Paragraph needs attention to what was published by cited authors. Author(14), In the article it was not possible to identify this aspect of the sentence. The summary mentions that sow terminates suckling when fights for teats occur.

2.2. Social integration within the herd

 Lines: 110-113. Jensen et al. (1991) [13] found that the percentage sucklings terminated by the sow gradually increases from less than 5% at day 1 post-partum to about 60% at day 10 post-partum, while Jensen (1988) [21] showed that this further increases towards week 4 post-partum.”

The same author in his publication of 1988 (21) had mentioned that the termination of suckling by sow was higher still in week 4 PP. And in a later publication of 1991 (13) mentions the above – to 60% at day 10PP.

This paragraph should be rewritten to better express this, taking into consideration the year of publication.

Lines: 116-117. Confusing sentence - Please check or elaborate better the idea of the sentence. ENGLISH LANGUAGE?

Suggestion: Between days 7 and 18 after farrowing the piglets attend with their mother for the first time the feeding sites [10, 18] and then continue to do this every morning.

 2.3. Slow and gradual weaning

Lines: 139-140. NOTE for editors: Please check the order of the years - if the Journal accepts this order as written or not.

 3. Effects of commercial weaning practices on piglet behavior and stress responses.

                Lines: 158-162. NEEDS TO BE EVALULATED for the actual idea of the authors  “piglets are typically weaned abruptly between 2.5 to 4.5 weeks of age.” – (2002). According to the EU directive 2008/120/EEC regarding weaning age, “No piglets shall be weaned from the sow at less than 28 days of age unless the welfare or health of the dam or the piglet would otherwise be adversely affected. Only if special conditions are offered the piglets can be weaned at 21 days of age.

                 Line: 163. It is clear

 Lines: 166-169. More recent publications on nutritional value of feeds to reduce stress would be helpful to understand the present conditions. And support the idea of this paragraph.

 Lines: 198-201. Piglets are fighting in order to impose dominance[37]. Ekkel et al. (1995) [38] and Pluske and Williams (1996) [39] demonstrated that piglets that are mixed at weaning show significantly more agnostic behaviors during the first hour after mixing than those that are not mixed at weaning.”

Suggestion: There are more recent articles on behaviours that can be added.

 Lines: From 201 to 237. – A very interesting read and information. The information is right from 1987 to 2016 and has been well presented. May be if possible, for certain interesting points expressed by earlier authors (1987; 1995; 1999) could possibly be backed up by more recent works if any.                

4. Socializing piglets during lactation in commercial settings

              Lines: 288-201. pro-mote social eating behavior while IS, a form of gradual weaning” Suggestion: In this sentence it would be good to use the full term of IS (intermittent suckling), as the abbreviation was only used in the beginning in the Introduction of the article.

Line: 371. In the conventional farrowing systemsor loose-housing systems the number….”SPACE

4.1.3. Social behavior of co-mingled piglets

 Lines: 434-436 – Ref Grimberg-Henrici et al. (2018) [112]. The housing system is not very clear in this sentence “in co-mingled piglets at the end of the lactation period compared with conventionally-reared piglets” According to the article the authors compared two housing systems during the lactational period – Group housing (sows+piglets) and Conventional single housing system. Using the term co-mingled piglets could imply that only the piglets were in groups.

 Line: 479 - exercise and not “exercize,” even in American English.

4.1.5. Effects of pre-weaning co-mingling on pre- and post-weaning mortality

Lines: 541-542 “The mortality rate of piglets in conventional housing systems where co-mingling of non-littermates is applied, seems to be lower and does not differ with conventional housing??? [98,91,96]” Please check this sentence. Appears like it is incomplete.

 4.2.3. Intermittent suckling and an extended lactation: effects on litter performance

         Table 3. Overview of different variants of intermittent suckling (IS).

Page 19 – Table 3 – Suggestion: Change position of presented authors to follow a sequence (a and then b).

Turpin et al. (2016b) [166]

Day 22

8 hours per day

7 days

29 days

Turpin et al. (2016a) [152]

Day 21 or Day 28

8 hours per day

7 days

 4.2.5. Intermittent suckling and a conventional lactation: impacts on the GIT, behavior and welfare – WELL WRITTEN & REVEWED – No comments

4.3. Gradual weaning in combination with co-mingling - WELL WRITTEN & DISCUSSED

 5. Conclusions

 Lines: 794-795 “In contrast to slow and gradual weaning under these circumstances, piglets are typically weaned abruptly between 2.5 to 4.5 weeks of age under commercial conditions”

Could this be confirmed? The European directive indicates 28 days as minimum which is equal to 4 weeks old or above. The mention of 2.5 weeks as minimum weaning age leads to the necessity for especial conditions which are not referred to in this review.

 Line: 814 - exercise and not “exercize,

English language is good. only a couple of corrections and the remark that it would ideal to to follow the English used in Europe as publication is in a Journal published in Europe.

Reviewer 2 Report

Comments for the authors

Major comments

-        you could discuss the economic impact/benefits of some of the discussed strategies for pig production, reporting the appropriate references.

-        Conclusions: you should use separate paragraphs to present the parts of this section.

Minor comments

-        L56: . the development of the natural..

-        L87: suckling piglets ??

-        L111: .. 5% on day 1 post-partum to about 60% on day 10 post-partum

-        L114: .. presents more opportunities..

-        L116: .. their mother for the first time feeding

-        L121: .. between weeks 2 and 7 post-partum

-        L163: It it clear that  ..

-        L226: .. mixing piglets cause stress..

-        L245: .. Yet, the effects of weaning

-        L257: .. in the intestinal mucosa

-        L295: .. congruent to the social adaptation

-        L382: .. sows with higher milk production

-        L404: .. leads to the reduced number

-        L445: .. It is after weaning that the

-        L456: .. was not a factor

-        L775: .. and then able..

-        L795: .. piglets have typically weaned

-        L818: The size of the group seems to depend 817 on the housing system; in multi-litter systems, groups consist mainly of four..

-        L823: .. are well-known impacts

-        L837: .. are potentially adaptable to a..

Minor comments

-        L56: . the development of the natural..

-        L87: suckling piglets ??

-        L111: .. 5% on day 1 post-partum to about 60% on day 10 post-partum

-        L114: .. presents more opportunities..

-        L116: .. their mother for the first time feeding

-        L121: .. between weeks 2 and 7 post-partum

-        L163: It it clear that  ..

-        L226: .. mixing piglets cause stress..

-        L245: .. Yet, the effects of weaning

-        L257: .. in the intestinal mucosa

-        L295: .. congruent to the social adaptation

-        L382: .. sows with higher milk production

-        L404: .. leads to the reduced number

-        L445: .. It is after weaning that the

-        L456: .. was not a factor

-        L775: .. and then able..

-        L795: .. piglets have typically weaned

-        L818: The size of the group seems to depend 817 on the housing system; in multi-litter systems, groups consist mainly of four..

-        L823: .. are well-known impacts

-        L837: .. are potentially adaptable to a..

Reviewer 3 Report

Reviewer Report

Reducing weaning stress in piglets by pre-weaning socialization and gradual separation from the sow: a review

I appreciate the opportunity to revise the present article which includes an extensive review of information related to weaning stress in piglets and the current strategies proposed to reduce the negative impacts that weaning has on animals. It is a highly relevant topic and, from an animal welfare perspective, discussing the methods applied to prevent negative states in animals is important. A general comment about this article would be that the organization of the topics and subtopics is a little confusing because inside each subtopic there is general and specific information, instead of presenting all the general concepts first. I hope my suggestions can help the authors.

Line 26: I consider that the information about the age of the piglets at weaning is more adequate for the abstract than for the simple summary. Please, consider adding in this section the average age of weaning in piglets.

Response:

Lines 46-48: The mixing with other piglets that are not from the same littler is another stressor that can cause aggression in weaned piglets (revise https://doi.org/10.1093%2Fjas%2Fskab311

Response:

Line 55: I suggest addressing some other consequences of weaning such as injuries to conspecifics, susceptibility to diseases due to cortisol’s immunocompromised, etc.

Response:

Lines 59-74: Please, try to summarize the aim of the article. As it is now, is very long and it is hard to understand what the real purpose of the review was, and which is part of the methodology that was followed.

Response:

Section 2, 2.1, and 2.2: I have the impression that the authors need to organize better these sections or improve the introductory paragraph of section 2. Lines 76-80 mention the eating behaviors of domestic pigs, then weaning behaviors that are similar to wild animals. Then in section 2.1, information about the farrowing process and maternal bonding is mentioned. I think it is relevant, but there is not a clear connection or relevance on why the review is starting with these topics. From my perspective, the article could start from section 3.

Response:

Line 88: Briefly mention why suckling attempts can be terminated. Also, mention what is the situation in the 10% of the piglets that are not mentioned in the manuscript.

Response:

Line 99: I recommend improving this section and mentioning more about maternal care (since it is included in the title of the subtopic). This article might be helpful: https://doi.org/10.3390/ani13030532

Response:

Line 110: Revise the journal’s guide for authors to amend in-text citation style throughout the manuscript.

Response:

Line 217: Similar to my previous comment, when mentioning cortisol, an association between cortisol increases and the consequences of this could help to understand the importance of preventing stress in piglets.

Response:

Line 228: Please, add a reference.

Response

Lines 246-248: This explanation about the release of cortisol and the activation of the HPA axis could be mentioned before since in previous sections cortisol is often mentioned but there was no mention of how the exposure to stressors is correlated to cortisol increase.

Response:

Section 4: Similar to my previous comment regarding section 2, it is a little confusing to follow the current organization of the subtopics. We go from farrowing to weaning to strategies before weaning, to general concepts about stress, then to weaning stress, and go back to lactation and then strategies before weaning. I would recommend organizing the article from general ideas/concepts, and then to comprise all the information about the weaning stress response, and then address the current strategies to avoid stress. Section 4 is way too long with too many subtopics. I suggest dividing this section into, at least, two ones to reduce the number of sub-topics and present the information in a more structured way.

Additionally, before the conclusions, the authors could propose some alternatives to improve the current strategies that are applied to piglets.

Response:

Conclusions: Generally, the conclusions can be an answer to the aim of the study. This section could be improved by summarizing the most relevant information. As it is now, a couple of the lines here seem to fit better in the introduction section. For example, at the end of the conclusion, it says “It also becomes clear that many factors can contribute to the success of these strategies, and that this may depend on farm-specific characteristics”. The conclusion that I expected to find was one where those “many factors” are explained and discussed according to the extensive literature search that the authors performed.

Response:

 Decision: Accepted with major changes.

Round 2

Reviewer 3 Report

I feel that the authors have adequately reviewed this manuscript and that it can now be published.